# ETS1 Protein Expression May Be Altered by the Complementarity of ETS1 mRNA Sequences with miR-203a-3p and miR-204-3p in Papillary Thyroid Carcinoma

**DOI:** 10.3390/ijms26031253

**Published:** 2025-01-31

**Authors:** Stefana Stojanović Novković, Sonja Šelemetjev, Jelena Janković Miljuš, Vladan Živaljević, Duško Dunđerović, Marija Milinković, Tijana Išić Denčić

**Affiliations:** 1Department of Endocrinology and Radioimmunology, Institute for the Application of Nuclear Energy—INEP, University of Belgrade, Banatska 31b, Zemun, 11080 Belgrade, Serbia; stefana@inep.co.rs (S.S.N.); sonja@inep.co.rs (S.Š.); jelenaj@inep.co.rs (J.J.M.); 2Faculty of Medicine, University of Belgrade, Doctor Subotić 8, 11000 Belgrade, Serbia; vladan.zivaljevic@med.bg.ac.rs; 3Clinic for Endocrine Surgery, University Clinical Center of Serbia, Pasterova 2, 11000 Belgrade, Serbia; 4Institute of Pathology, Faculty of Medicine, University of Belgrade, Doctor Subotić 1, 11000 Belgrade, Serbia; drdundjerovic@gmail.com; 5Department of Pathology, Clinical Center of Serbia, Pasterova 2, 11000 Belgrade, Serbia; marija.ksc.path@gmail.com

**Keywords:** PTC, miRNA, cancer biomarker, prognostic factor, microRNA, subcellular localization

## Abstract

The expressions of ETS1, miR-203a-3p, and miR-204-3p in papillary thyroid carcinoma (PTC) are poorly described, and their clinical significance is unclear. To determine the prognostic value of ETS1 (E26 transformation-specific), its levels in divergent cell compartments were paired with miR-203a-3p/-204-3p levels and linked to the presence of unfavorable clinical characteristics of PTC patients. Immunohistochemistry and Western blot were performed to evaluate ETS1 protein expression in PTC and matched nonmalignant thyroid tissue (NMT). qPCR was utilized to quantify ETS1 mRNA, miR-203a-3p, and miR-204-3p expressions. Bioinformatic analysis was applied to predict biological interactions. Although there was a significant increase in ETS1 protein expression (*p* < 0.05), no difference was observed in ETS1 mRNA levels between PTC and matched NMT (*p* > 0.05). 98.7% of PTC samples exhibited positive staining for the ETS1 protein, detected in the nucleus, the cytoplasm, or both. In contrast, the ETS1 protein had positive staining in 70.9% of NMT samples, primarily localized in the nucleus. ETS1 cytoplasmic levels correlated with the pT status of PTC patients (*p* = 0.020, r = −0.267), while nuclear levels correlated with the occurrence of lymph node metastasis (*p* = 0.020, r = −0.271). According to the bioinformatic analysis, the 3′-untranslated region of ETS1 mRNA shares a seed sequence with miR-203a-3p/-204-3p. The mutual distribution of ETS1 and miR-203a-3p levels differs between aggressive and non-aggressive PTCs. ETS1 could be used in the identification of high-risk PTC patients; however, its subcellular localization should be considered. PTC aggression could be influenced by increased cytoplasmic ETS1 protein levels, which may be affected by reduced levels of miR-203a-3p or miR-204-3p.

## 1. Introduction

Papillary thyroid carcinoma (PTC), the most prevalent endocrine tumor with a growing incidence in recent decades [1], displays significant variations in histopathological features and clinical outcomes. Thyroid suppression treatment, radioiodine ablation, or surgery can effectively treat the majority of PTC cases; nevertheless, 5% to 20% of patients experience tumor recurrence, and if distant tumor recurrences occur, the 10-year survival rate drops to about 40% [2,3]. Therefore, there is a constant challenge in the diagnosis and treatment of PTC. The development of molecular determinants of disease aggressiveness level may enhance clinical management of PTC patients by allowing more aggressive treatments to be administered only to those patients who are more likely to recur.

ETS1 (E26 transformation-specific) transcription factor belongs to the ETS family of proteins that is involved in a number of cellular functions, such as differentiation, apoptosis, and proliferation [4]. Additionally, ETS1 has been linked to angiogenesis in pathological settings, including the establishment and growth of tumors [5]. It plays an important role in cancer progression due to its ability to induce the transcription of genes related to invasion, angiogenesis, and metastasis. Although the ETS1 protein is normally found in the nucleus, it has also been found in the cytoplasm of endothelial cells during angiogenesis, including tumor vascularization, in the cytoplasm of tumor cells of various origins, and in the cytoplasm of tumor stromal fibroblasts [6,7,8,9]. In colon cancer, ETS1 expression is directly related to the malignancy, with no expression in normal tissues and the highest expression in carcinomas with lymph node metastasis [10]. In prostate and breast cancer, it is a marker of poor prognosis [11,12], while ETS1 is downregulated in poorly differentiated pancreatic adenocarcinoma [7].

Although aberrant ETS1 activation has been documented in various solid tumors [5,6,7,8,9,10,11,12], a thorough and systematic analysis elucidating the role of ETS1 in thyroid tumor progression remains limited. Its expression in PTC is poorly characterized, and the clinical significance for PTC patients is ambiguous. While Fuhrer et al. [13] found elevated levels of ETS1 in PTC when compared to benign thyroid nodules and normal thyroid tissues, Nakayama et al. [14] observed strong expression of ETS1 in the majority of malignant thyroid tumors, a minority of benign thyroid tumors, and no expression in normal thyroid follicular cells. The TCGA study revealed ETS1 as a differentially expressed gene between the lower and higher stages of PTC patients [15], and the transcriptional activity of ETS1 and ETS2 was found to be crucial for the transformation of thyroid follicular cells [16]. ETS1 mRNA expression was found to be upregulated in PTCs harboring the oncogene BRAFV600E mutation [17,18], and the mutation itself caused an increase in ETS1 protein expression and its nuclear accumulation [17]. Subsequent investigation revealed the direct role of ETS1 as a transcriptional regulator of toll-like receptor 4 (TLR4) expression downstream of MAPK/ERK signaling that is triggered by the presence of the BRAFV600E mutation [17]. Furthermore, the study of Song et al. [19] provided transcriptomic insights that the interaction between the BRAFV600E and TERT promoter mutations is mediated by ETS factors in PTC. Additionally, ETS1 was found to be a genome-wide effector of RAS/ERK signaling in epithelial cells [20]. Conversely, overexpression of the lncRNA SLC26A4-AS1 enhanced autophagy mediated by ITPR1 (the inositol 1,4,5-trisphosphate receptor type 1) and prevented the growth and progression of PTC by recruiting ETS1 [21,22].

MicroRNAs (miRs) are non-coding, single-stranded RNA molecules composed of approximately 22 nucleotides that have remained conserved throughout evolution. MiRs serve multiple functions, and their regulation mechanism is currently debated [23]. They have been proven to regulate gene expression post-transcriptionally and have been identified as crucial regulators in cancer biology [24,25]. They typically complementarily bind to the 3′ untranslated region (3′ UTR) of targeted mRNAs, inhibiting protein translation and/or degrading the target mRNA. MiR-203a-3p and miR-204-3p have been shown to exhibit differential expression patterns in various cancers, including thyroid carcinoma, but their function in thyroid carcinomas is still doubtful [26,27,28,29,30,31]. In lung adenocarcinoma, ETS1 was validated as a downstream target of miR-204-3p [32]. Bioinformatic predictions indicate that miR-203a-3p and miR-204-3p share a seed sequence with the 3’UTR of ETS1 mRNA, but there have been no studies on mutual expression of ETS1 and miR-203a-3p/204-3p in thyroid carcinomas.

In this study, ETS1 expression was evaluated in PTC patients to determine its prognostic significance. In the first part of this study, ETS1 protein levels in divergent cell compartments were associated with known clinicopathological parameters of PTC patients. Afterwards, cytoplasmic ETS1 levels were paired with miR-203a-3p/204-3p high/low levels, and the results were linked to the incidence of unfavorable clinical characteristics of the patients.

Exploration of the relation between ETS1, miR-203a-3p, and miR-204-3p in the context of PTC is not only significant for elucidating the molecular mechanisms of thyroid carcinogenesis but also for developing targeted therapeutic strategies. Understanding the molecular mechanisms by which these miRs influence the expression of the ETS1 protein may help identify new therapeutic targets for the management of PTC, especially in cases where traditional treatments are ineffective.

## 2. Results

### 2.1. ETS1 Expression in PTC and Matched NMT

There is a statistically significant difference between the median values of the ETS1 IHC score between PTC and matched NMT (Related-samples Wilcoxon signed rank test, *p* = 0.000), and there is a statistically significant difference in the distribution of the values of the ETS1 IHC score in PTC and matched NMT (Figure 1a, related-samples Friedman’s two-way analysis of variance by ranks, *p* = 0.000).

The application of WB methodology yielded similar results. Related-samples Wilcoxon signed-rank test showed a statistically significant difference between the median values of ETS1 expression in PTC vs. its median values in matched NMT, tested by WB (*p* = 0.007), while related-samples Friedman’s two-way analysis of variance by ranks showed a statistically significant difference in the distribution of the values of ETS1 expression in PTC and matched NMT, gained by WB application (Figure 1b, *p* = 0.046).

Therefore, there is a statistically significant difference in the levels of the ETS1 protein expressed in PTC vs. its expression in matched NMT (Figure 1a,b, *p* < 0.05).

On the other hand, there is no statistically significant difference in the level of expression of ETS1 mRNA between PTC and NMT (Related-samples Wilcoxon signed-rank test, *p* = 0.887, Figure 1c).

### 2.2. Bioinformatic Analysis and Model Prediction

According to bioinformatic analysis and model prediction, miR-203a-3p may interact with ETS1 mRNA in two complementary positions: starting positions of seed locations 1343 and 2350 (target score of 63%), while miR-204-3p may interact with ETS1 mRNA in two additional positions: starting positions of seed locations 1949 and 2802 (target score of 98%) (Table 1 and Figure 2).

Given that both methods for testing protein expressions (IHC and WB) revealed higher ETS1 protein expression in PTC than in matched NMT, and that qPCR results revealed no difference in ETS1 mRNA expression between PTC and matched NMT, it is assumed that the ETS1 protein expression is regulated posttranscriptionally. In our previous work we have shown that miR-203a-3p and miR-204-3p are the two miRs with generally downregulated expression in PTC compared to its expression in NMT [28]. Since bioinformatics analysis and model prediction revealed that miR-203a-3p and miR-204-3p might complementary bind ETS1 mRNA, it is possible that the presence of these miRs is the cause of reduced translation of ETS1 in PTC cases, particularly since mature miR could be found only in the cytoplasm of the cells. In other words, lower expression of miR-203a-3p and miR-204-3p in PTC (compared to matched NMT) might be the cause of increased ETS1 protein expression in the cytoplasm of PTC cells.

To further evaluate this possibility, we analyzed in detail the subcellular localization of the ETS1 protein in PTC and matched NMT samples.

### 2.3. The Level of Expression of the ETS1 Protein by Cell Compartments

The subcellular location of ETS1 and the degree of expression of the ETS1 protein in each cell compartment were assessed using immunohistochemistry (IHC). There was a difference in the distribution of total (Figure 3e,f; Wilcoxon signed-rank test, *p* = 0.000) and cytoplasmic (Figure 3c,d; Wilcoxon signed-rank test, *p* = 0.000) ETS1 IHC staining between PTC and NMT, but there was no difference in nuclear ETS1 IHC staining between PTC and NMT (Figure 3a,b; Wilcoxon signed-rank test, *p* = 0.056).

PTC showed predominantly positive ETS1 IHC staining, either in the nucleus or in the cytoplasm, or in both (76/77, 98.7%). ETS1 was found in the cytoplasm of 92.2% (71/77) of PTC cases and in the nucleus of 81.8% (63/77) of PTC cases, at variable levels (Figure 3b,d,e). There was no correlation of the ETS1 IHC score in the cytoplasm with the ETS1 IHC score in the nucleus of PTC cells (Figure 3h, *p* = 0.627). There was no statistically significant difference in the distribution of cytoplasmic or total ETS1 IHC staining among divergent PTC subtypes (Figure 4b, *p* > 0.05; see Appendix A). But there was a difference in nuclear ETS1 staining among divergent PTC subtypes (Kruskal-Wallis Test, *p* = 0.017, and the Median Test, *p* = 0.016). This difference is predominately caused by the higher nuclear ETS1 staining scores of the follicular variant of PTC (for details see Appendix A, Pairwise Comparisons of PTC subtypes). All other comparisons among PTC subtypes did not gain statistically significant differences in the expression of ETS1 in the nucleus (*p* > 0.05, Appendix A). Representative micrographs are shown in Figure 5.

On the contrary, among 55 NMT adjacent to the PTC, the ETS1 protein exhibited positive staining in 39 (70.9%) cases with predominately nuclear localization (Figure 3a,c). ETS1 was found in the nucleus of 39/55, 70.9% of cases, at variable levels, and in the cytoplasm of 17/55, 30.9% of cases, with predominately weak staining. It is interesting to notice that, if positive, thyroid cells of the follicles close to the tumor showed intense nuclear staining, while the intensity of ETS1 IHC staining of NMT declined along with the distance from the tumor. The distribution of expression of ETS1 is dependent on the thyroid nonmalignant neoplasia subtype (Figure 4a, Appendix A). This difference is predominately caused by the lower ETS1 staining scores in both cell compartments of normal (healthy) thyroid tissue compared to ETS1 expression in divergent nonmalignant thyroid lesions (*p* < 0.05; for details see Appendix A, Pairwise Comparisons of NMT subtypes). Normal (healthy) thyroid tissue had no to low ETS1 expression, while ETS1 could be detected at variable levels in Hashimoto’s thyroiditis, nodular goiter, and thyroid adenomas. Cases with Hashimoto’s thyroiditis and adenomas, if positive, showed predominately nuclear staining. There was no difference in the expression of ETS1 among nodular goiter, thyroiditis, and thyroid adenoma (*p* > 0.05, Appendix A). Representative micrographs are shown in Figure 5.

### 2.4. Association of ETS1 IHC Staining and Clinicopathological Data of PTC Patients

The correlation of ETS1 IHC staining and clinicopathological data of PTC patients is shown in Table 2. As it could be seen, cytoplasmic levels of the ETS1 protein correlated with the pT status of PTC patients (*p* = 0.020, r = −0.267), while the nuclear levels of ETS1 correlated with the lymph node metastasis of the patients (*p* = 0.020, r = −0.271). This means that in PTC, lower values of ETS1 in the cytoplasm correlate with a higher pT grade of the tumor, while lower values of ETS1 in the nucleus correlate with the lnm occurrence.

To test if ETS1 could predict pT grade or lnm occurrence, we performed ROC analysis and plotted the ROC curve (Figure 6). ROC analysis confirmed a statistically significant difference between cytoplasmic levels of the ETS1 protein in PTCs with pT grade 1-2 vs. pT grade 3-4 (Figure 6a, AUC = 0.689, SE = 0.063, *p* = 0.007). With the cutoff set at 1.75 for cytoplasmic levels of ETS1, the sensitivity is 65.3%, and the specificity is 70.4%. ROC analysis also confirmed that nuclear levels of ETS1 could predict the presence of lnm (Figure 6b, AUC = 0.704, SE = 0.072, *p* = 0.021), and with the cutoff set at 0.18 for nuclear staining of ETS1, the sensitivity is 75.8% and the specificity is 61.5%.

### 2.5. Association of ETS1 IHC Staining, miR-203a-3p/miR-204-3p Level of Expression, and the Clinicopathological Data of PTC Patients

For further testing the possible influence of the amount of tested miRs on the ETS1 protein expression and whether this relationship has any influence on the unfavorable factor occurrence, we divided our cohort into high and low miR-expressing groups, crosslinked it with the patients’ clinicopathologic data, and determined the median ETS1 values in each category. The results are shown in Figure 7.

As it could be seen, the distribution of ETS1 median across miR-203a-3p low/high groups differs between aggressive and non-aggressive PTCs (Figure 7). In a group of PTC patients with more aggressive clinical features (higher PTC grade, invasion or higher levels of tumor infiltration), ETS1 levels were higher in the miR-203a-3p high expressed group than in the miR-203a-3p low expressing one, whereas in non-aggressive PTCs, median ETS1 expression was lower (or equal) in the miR-203a-3p high expressed group than in the miR-203a-3p low expressing group.

We found no such relation between miR-204-3p expression levels, ETS1 protein expression, and clinicopathological data tested.

## 3. Discussion

Thyroid carcinoma represents a significant area of research due to its increasing incidence in the last decade and the complexity of its molecular background [1,33]. About 85% of thyroid cancer patients have papillary thyroid carcinomas (PTCs), which are frequent endocrine tumors with a relatively low mortality rate but with a high probability of persistence or recurrence [3]. While the general prognosis of PTC patients is well, the survival in the recurrent group of patients is significantly lower [2]. With a better understanding of the molecular processes involved in thyroid tumorigenesis, the use of molecular techniques has become particularly attractive for improving differential diagnosis of thyroid nodules, enabling disease prognosis and perhaps adjusting treatment approaches.

Among the various molecular players involved in the pathogenesis of PTC, the transcription factor ETS1 has garnered attention because of its roles in proliferation, differentiation, and apoptosis, the cell processes critical to cancer development, progression, and metastasis [4,5,8,13,14,16,34]. According to our results, the ETS1 protein is overexpressed in PTC compared to adjacent NMT (*p* < 0.05), and this goes in line with the previously published data that total cell amount of ETS1 is raised in malignant cells [2,5,14,16]. 98.7% of PTC cases included in this study showed positive ETS1 IHC staining, either in the nucleus or in the cytoplasm, or in both. Similarly, 97.9% of PTCs showed positive staining of ETS1 in a study of Nakayama T et al. [14], and eight- to ten-fold upregulation of ETS1 was observed in PTC compared with benign thyroid nodules and normal thyroid tissues in the study of Fuhrer D et al. [13]. But, while there have been a few studies on ETS1 protein expression in PTC [13,14], none of them have examined ETS1 expression in distinct cell compartments or in PTC subtypes. Our findings showed a difference in the subcellular distribution of ETS1 between neighboring NMT and PTC (*p* < 0.05). While there was no difference in nuclear expression of ETS1 between PTC and NMT (*p* > 0.05), the cytoplasmic levels of ETS1 presented divergent distribution between PTC and NMT (*p* < 0.05). Therefore, according to our study, the rise in total ETS1 levels in PTC is predominantly caused by the increased levels of the ETS1 protein in the cytoplasm of malignantly transformed thyroid cells. However, Peyret et al.’s investigation [17] observed that although ETS1 was present in both the cytoplasm and the nucleus, the BRAFV600E mutation led to increased ETS1 protein expression and accumulation in the nucleus of thyroid cells that had undergone malignant transformation. Song et al. [19] have proposed a possible mechanism for the phenomenon of the synergistic effects of the two most common mutations in thyroid carcinomas on the poor clinical outcomes in PTC. They discovered that ETS factors induced by BRAFV600E selectively bound to the mutant TERT promoter, which in turn affects PTC invasiveness and progression in a synergistic manner.

The functional involvement of ETS proteins in regulating apoptosis has been the subject of numerous investigations [16,35,36,37]. According to the findings of de Nigris F. et al. [16], programmed cell death was induced when ETS1 activity was suppressed. The crucial function of the ETS1 protein in T-cell survival was demonstrated by disruption of the *Ets-1* gene [35]. Furthermore, it was demonstrated that the p42 splice variant of ETS1 can cause human colon cancer cells to undergo apoptosis [36], and that the survival of thyroid carcinoma cells, but not normal thyroid cell lines, depends on the activity of the ETS1 and ETS2 proteins [16]. All the carcinoma samples and the evaluated human thyroid carcinoma cell lines showed elevated ETS activity, which was necessary to preserve the neoplastic phenotype of the human thyroid carcinoma cell lines. Furthermore, Nigris F. et al. [16] reported that there was no difference in the expression of the ETS gene family between normal thyroid and thyroid benign adenomas. On the contrary, according to our study, there was a significant difference in the expression of ETS1 between normal (healthy) thyroid tissue and tested nonmalignant thyroid neoplasia (*p* < 0.05) as ETS1 was expressed at variable levels in Hashimoto’s thyroiditis, nodular goiter, and thyroid adenomas, while normal (healthy) thyroid tissue presented no to low ETS1 expression. It is interesting to notice that, in our series of nonmalignant thyroid samples, if positive, nonmalignant thyroid cells of the follicles close to the tumor showed intense nuclear staining, while the intensity of ETS1 IHC staining of NMT declined along with the distance from the tumor. One possible explanation for this observation is that some of the genes that respond to ETS1 code for specific proteases, including urokinase-type plasminogen activator [38,39,40,41,42] and matrix metalloproteinases MMP-1 and MMP-9, which modify the extracellular matrix, making it easier for tumor cells to invade and increasing their capacity to spread. ETS1 and MMP-1 are co-expressed in skin angiosarcoma [41], whereas ETS1 is co-expressed with MMP-1 and MMP-9 in ovarian carcinoma cells and in stromal fibroblasts of breast carcinoma [43]. Since invasive behavior is part of the endothelium activation program, ETS1 may be responsible for activating the required proteases [44]. ETS1 appears to play a role in regulating the invasive behavior of many normal and tumor-like cells, as it has been shown that ETS1 is required for adopting an angiogenic, blood vessel-forming phenotype in endothelial cells [42,45,46,47]. Furthermore, it was shown that the expression of ETS1 positively correlates with factors of angiogenesis such as angiopoietin-2 and vascular endothelial growth factor (VEGF), hence stimulating the formation of new blood vessels [44,46,47]. Therefore, in the carcinomas of epithelial origin, ETS1 might have a dual function; it may induce tumor vascularization and consequently provide oxygen and nutrition to cancer cells, and it may enhance invasion by the activation of ECM-degrading proteases in the cancer and/or in stromal cells. Furthermore, it may be that the ETS1 protein controls divergent pathways in stromal and epithelial tumor compartments, and the crosstalk between these two is essential for the development of cancer aggressiveness.

Many studies correlated the level of expression of the ETS1 protein with the occurrence of unfavorable factors of diverse carcinoma types, but the gained results are controversial. In pancreatic carcinomas, ETS1 expression has no association with the lymph node metastasis, tumor size, prognosis, or tumor-node-metastasis stage [2], while in colorectal carcinoma, it correlated with the lymph node metastasis, the depth grading of tumor invasion, as well as its invasion into the lymphatic or venous vessel [9]. In gastric cancer, ETS1 expression correlated significantly only with the presence of lymph node metastasis [6]. In our study on thyroid carcinoma, lower levels of ETS1 in the cytoplasm correlated with higher pT status of PTC patients, and lower nuclear levels of ETS1 correlated with the occurrence of metastasis in the local lymph nodes, while no significant correlation was found of total ETS1 amount with any of the tested unfavorable clinicopathological PTC factors. As far as we know, none of the studies published to date have correlated ETS1 expression with unfavorable clinicopathological information of PTC patients by cell compartments.

While ETS1 protein levels differed between PTC and NMT in terms of quantity and subcellular distribution, our findings show that ETS1 mRNA was expressed at similar levels in both. MiRs are small non-coding RNAs that regulate gene expression post-transcriptionally and have been recognized as key regulators in cancer biology. Because of its suppressive function, a specific regulatory mechanism involving miR may result in the overexpression of oncogenes and/or downregulation of tumor suppressor genes [23,24,25]. Alterations in miR expression have been reported as significant regulators of thyroid cancer occurrence, development, and progression [24,25,28,48,49]. MiR-203a-3p and miR-204-3p are recently discovered miRs with still not fully explained biological function in PTC. The dysregulation of these miRs has been associated with tumor growth, invasion, and metastasis, suggesting that they may serve as potential biomarkers and therapeutic targets in PTC. For instance, miR-203a-3p has been reported to suppress cell proliferation and migration by targeting specific oncogenes, thereby acting as a tumor suppressor [50,51]. Similarly, miR-204-3p has been implicated in the regulation of pathways that govern cell cycle progression and apoptosis, further underscoring its role in the pathogenesis of thyroid carcinomas [26,27]. As previously stated, the primary mechanism by which miRs act is translation suppression, which is induced by complementary binding of miR to the 3′-UTR of mRNA, and the primary site of action for miRs is the cell cytoplasm, as mature miRs are found there. Furthermore, it was previously shown that the expression of miR-203a-3p and miR-204-3p is downregulated in PTC compared to NMT [26,27,30], and ETS1 was shown to be a downstream target of miR-204-3p in lung adenocarcinoma [32]. In this work, bioinformatics analysis showed that miR-203a-3p and miR-204-3p target ETS1 with relatively high target scores. Therefore, it is possible that while ETS1 mRNA expression remains unaffected in PTC compared to NMT, in NMT, miR-203a-3p/miR-204-3p complementary binds to ETS1 mRNA in the cytoplasm of thyrocytes, which restricts ETS1 protein translation. On the other hand, ETS1 mRNA translation into the ETS1 protein is unrestricted in the cytoplasm of PTC cells due to the lower amount (or absence) of miR-203a-3p and miR-204-3p. This goes in line with our observation of elevated levels of the ETS1 protein only in the cytoplasm of PTC cells, compared to its expression in adjacent NMT, and no change in its nuclear expression. Many previously published data confirmed the presence of ETS1 mRNA in cancer tissue [2,6,9,13], but we have not found the manuscript describing the association of down-regulation of cytoplasmic expression of the ETS1 protein, miR-203a-3p/miR-204-3p expression, and unfavorable clinicopathological factor occurrence.

In addition to their roles in tumor biology, the expression levels of ETS1, miR-203a-3p, and miR-204-3p may serve as prognostic indicators in PTC. Studies have shown that patients with higher levels of these mRs tend to exhibit more aggressive disease features, including increased rates of metastasis and recurrence [26,28]. In this study, the opposite relations of ETS1 and miR-203a-3p levels were noticed in low- and high-risk groups of PTC patients. In a group of PTC patients with non-aggressive clinical features with low miR-203a-3p levels, the levels of the ETS1 protein were high. On the other hand, in aggressive tumors (PTCs of the highest level of infiltration, pT or pTNM stage), ETS1 levels were higher in the miR-203a-3p high-expressing group than in the miR-203a-3p low-expressing group. This could imply that miR-203a-3p and ETS1 may interact, and while they are in balance, PTC has a non-aggressive flow, and the disruption of the balance may turn PTC to aggressive behavior. But it should be kept in mind that some other complex interactions may additionally occur and subsequently influence the process of PTC spreading. In our previous study we have shown that the presence of some other factors, such as long non-coding RNA or mutation, may furthermore influence aggressive behavior of PTC [29]. More precisely, long non-coding RNA BANCR may act as a sponge for miR-203a-3p and miR-204-3p in PTC, further reducing their levels, but the interactions appear to be dependent on the presence of the BRAFV600E mutation [29]. These findings help us understand the underlying molecular pathways that may impact PTC’s aggressive behavior. This suggests a multifaceted regulatory network where ETS1, miR-203a-3p, and miR-204-3p interact with other molecular pathways, potentially leading to enhanced tumor aggressiveness and poor clinical outcomes in PTC patients. This highlights the potential utility of ETS1, miR-203a-3p, and miR-204-3p as biomarkers for risk stratification in PTC, which could guide clinical decision-making and therapeutic interventions. The exploration of the connection between ETS1 and miR-203a-3p/miR-204-3p in the context of PTC is not only significant for elucidating the molecular mechanisms of thyroid carcinogenesis but also for developing targeted therapeutic strategies. Understanding the molecular mechanisms underlying the regulation of ETS1 by these miRs could unveil novel therapeutic targets for the treatment of PTC, particularly in cases that exhibit resistance to conventional therapies.

Despite several limitations in our study, including a lack of cell culture experimental confirmation that miR-203a-3p and miR-204-3p could directly regulate ETS1 protein expression, a relatively small sample size, and no follow-up data, it is reasonable to believe that ETS1 activity is context dependent. The relative amount of miR-203a-3p and miR-204-3p in the tissue may account for ETS1 protein quantity and tissue-specific effects. As research continues to unravel the complexities of these interactions, it is imperative to consider the broader implications for patient management and treatment outcomes in thyroid cancer. The integration of molecular profiling into clinical practice may ultimately enhance our ability to personalize treatment approaches and improve prognostic accuracy for PTC patients.

In summary, the relationship between ETS1 and miR-203a-3p/miR-204-3p is a promising area of investigation in the field of thyroid oncology. By elucidating the mechanisms through which these molecules interact and influence tumor behavior, researchers may pave the way for novel diagnostic and therapeutic strategies that could significantly impact PTC patient management. Future studies are warranted to further explore this intricate network and its implications for understanding thyroid cancer and treating thyroid carcinoma patients.

## 4. Materials and Methods

### 4.1. Patients and Tissue Samples

The study was performed in accordance with the Declaration of Helsinki and obtained approval from the Ethics Committee of the Faculty of Medicine, University Clinical Center of Serbia. Paired samples of PTC tumors and their corresponding adjacent normal thyroid tissues were obtained from patients who underwent routine surgical resection at the Clinic for Endocrine Surgery, Clinical Center of Serbia, Belgrade. Tissue samples were promptly frozen in liquid nitrogen after resection and stored at −80 °C for subsequent RNA and protein analysis. Simultaneously, tissue specimens from the same patients were preserved in formalin and embedded in paraffin and subsequently utilized for immunohistochemical analysis. All participants provided informed consent for the use of their biological material in this study. Histopathological diagnoses were determined by the consensus opinion of two pathologists based on standard cytohistopathological guidelines [52].

Clinicopathological characteristics, such as patient age and gender, size of the tumor, extrathyroidal extension (Ei), intraglandular dissemination (ID), lymph node metastasis (Lnm), and distant metastasis status, were gathered through a review of pathology reports. Carcinomas were classified based on the pathological tumor-node-metastasis (pTNM) staging system as defined by the American Joint Committee on Cancer (AJCC) [53]. Moreover, each case was re-evaluated for the degree of tumor infiltration (DTI) according to the previously established classification: 1—completely encapsulated tumors; 2—non-encapsulated tumors without invasion of the thyroid capsule; 3—tumors with invasion of the thyroid capsule; and 4—tumors exhibiting extrathyroidal extension (Ei) [54].

In this study, 77 PTC samples were analyzed using various methods. For immunohistochemical (IHC) analysis, all 77 PTC tissue sections were examined, with adjacent nonmalignant tissue (NMT) identified in 55 samples. The PTC cohort consisted of 14 cases of the classical PTC variant (PTC cv), 31 cases of the follicular variant (PTC fv), 13 cases exhibiting mixed histopathological features, combining follicular and classical areas in varying proportions (PTC mix), and 19 cases with rare PTC histologic subtypes (PTC rare), including solid, Warthin-like, tall cell, clear cell, and oxyphilic variants. The NMT cohort consisted of 24 normal (healthy, pathohistologically untransformed) thyroid tissues, 11 thyroid adenomas, 13 nodular goiters, and 7 cases of thyroid thyroiditis. Western blot analysis was performed on 34 PTC samples, 16 of which had matched NMT. qPCR for ETS1 mRNA expression was conducted on 53 PTC samples and 43 matched NMT, while qPCR for miR-203a-3p and miR-204-3p expression was performed on all 77 PTC cases. The differences in sample numbers for each analysis reflect the availability of tissue required for specific methods.

The patient cohort included 59 females (76.6%) and 18 males (23.4%), with ages at diagnosis varying from 14 to 89 years (mean ± SD = 53.61 ± 15.92 years). Tumor sizes varied between 2 mm and 120 mm (mean ± SD = 30.70 ± 17.63 mm). Intrathyroidal dissemination (ID) was identified in 40 cases (51.9%), extrathyroidal invasion (Ei) in 20 cases (26.0%), and lymph node metastasis (Lnm) in 13 cases (16.9%). In this study, no patients had documented distant metastases.

### 4.2. RNA Isolation

Frozen PTC tissue samples (approximately 100 mg) were homogenized using the TissueLyser LT (Qiagen GmbH, Hilden, Germany) with TRIzol reagent (Invitrogen, Carlsbad, CA, USA) at a frequency of 50 Hz for 5 min. RNA extraction was performed according to the manufacturer’s guidelines. The concentration of RNA was measured with an Epoch microplate spectrophotometer (BioTek, Winooski, VT, USA).

### 4.3. Reverse Transcription and Quantitative PCR Analysis for miR-203a-3p and miR-204-3p

Reverse transcription and quantitative real-time PCR for miR-203a-3p and miR-204-3p were carried out according to the methodology described in our earlier publication [28]. The sequences of the stem-loop primers used in the reverse transcription reaction are listed below:miR-203a-3p-RT: 5′-GTCGTATCCAGTGCAGGGTCCGAGGTATTCGCACTGGATACGACCTAGTG-3′miR-204-3p-RT: 5′-GTCGTATCCAGTGCAGGGTCCGAGGTATTCGCACTGGATACGACACGTCC-3′miR-u6-RT: 5′-GTCGTATCCAGTGCAGGGTCCGAGGTATTCGCACTGGATACGACAAAAATATGG-3′

Additionally, the primers used for qPCR are as follows:miR-U6-Fw: 5′-GCGGTCGCAAGGATGACACG-3′miR-203a-3p-Fw: 5′-CGGCGGTGTGAAATGTTTAGGAC-3′miR-204-3p-Fw: 5′-GCGGTGCUGGGAAGGCAAAG-3′universal miR-Rv: 5′-CCAGTGCAGGGTCCGAGGTAT-3′

### 4.4. Reverse Transcription and Quantitative PCR for ETS1

A total of 1 μg of isolated RNA was reverse transcribed using the iScript Select cDNA Synthesis Kit (Bio-Rad Laboratories, Inc., Hercules, CA, USA) with random hexamer primers for cDNA synthesis. The reaction was carried out with the following temperature conditions: 5 min at 25 °C, 30 min at 42 °C, and 5 min at 85 °C.

Quantification of complementary DNA was performed via real-time PCR with SsoAdvanced Universal SYBR Green Supermix (Bio-Rad Laboratories, Inc., Hercules, CA, USA). The expression level of the *ETS1* gene was normalized to *GAPDH* expression using the 2^−ΔΔCt^ method, representing the relative expression of ETS1. Each reaction was conducted in triplicate and repeated at least twice. The qPCR cycling conditions were as follows: pre-incubation for 2′ at 50 °C, denaturation for 10′ at 95 °C, followed by 40 amplification cycles consisting of 15″ at 95 °C and 1’ at 60 °C. The melting curve analysis included 15″ denaturation at 95 °C, 60″ of annealing at 60 °C, and a final 30″ denaturation at 95 °C. All reactions were performed using a 7500 real-time PCR system.

The following primer sequences were used for qPCR:*GAPDH*-Fw: 5′-GAAGGTGAAGGTCGGAGT-3′*GAPDH*-Rv: 5′-GAA GATGGTGATGGGATTTC-3′*ETS1*-Fw: 5′-GCCCAGCTTCATCACAGAGT-3′*ETS1*-Rv: 5′-CCCCGAGTTTACCACGACTG-3′

### 4.5. Protein Extraction and Western Immunoblotting

Postoperative tissue samples were used to isolate total proteins. 1 ml of cold homogenization buffer (20 mM Tris HCL, 137 mM NaCl, 10% glycerol, 1% NP-40, 2 mM EDTA), containing 10 µl of protease inhibitor cocktail, was added to 100 mg of tissue. The tissue was homogenized for 10′ at 50 Hz, followed by centrifugation at 12,000 rpm for 10′ at 4 °C. The supernatant was collected, aliquoted, and used for the determination of protein concentration. Protein concentration was measured using the BCA protein assay kit (Pierce, Rockford, IL, USA). For Western blot analysis, 100 µg of protein was used.

Tissue homogenates were loaded onto a 10% polyacrylamide gel for separation, followed by transfer to a nitrocellulose membrane (Amersham Protran, GE Healthcare Life Sciences, MA, USA). The membranes were incubated overnight at 4 °C with anti-ETS1 monoclonal antibody (clone JM92-32, Cat #MA5-32732, Thermo Fisher Scientific, Rockford, IL, USA) and anti-β-actin antibody (clone AC-15, Cat #MA1-91399, RRID: AB_2273656, Thermo Fisher Scientific, Rockford, IL, USA). The dilution for both antibodies was 1:1000. Secondary antibodies were applied for 45′ at room temperature with a dilution of 1:2000.

Protein detection was carried out using an ECL substrate (Thermo Fisher Scientific, Rockford, IL, USA). Densitometric analysis was performed using ImageLab 6.1. software (BioRad Laboratories, Hercules, CA, USA). β-actin was used as the endogenous control for normalization of the bands, along with an additional internal control (the protein homogenate used in each blot to normalize the results across different experiments).

### 4.6. Immunohistochemistry

Immunohistochemical staining was conducted using the same antibody as for Western blot (anti-ETS1 monoclonal antibody, clone JM92-32, Cat #MA5-32732, Thermo Fisher Scientific, Rockford, IL, USA) in a 1:100 dilution. Deparaffinization was performed using xylene, followed by rehydration in a series of ethanol solutions with increasing concentrations. Hydrogen peroxide was used to block endogenous peroxidase activity. To prevent non-specific binding, slides were incubated with normal horse serum for 30 min at room temperature. Before the incubation with the primary antibody, an antigen retrieval step was performed by immersing the slides in a sodium citrate solution (pH 6.0) and heating them to boil in a microwave for 30′. Once cooled to room temperature, the slides were washed three times with washing buffer (PBS). They were then incubated with the primary antibody overnight at 4 °C. Incubation with the secondary antibody was performed at room temperature for 45’. The dilution of secondary antibody was 1:200. Vectastain ABC kit (PK-6100, Vector Laboratories, Newark, CA, USA) was used for signal amplification, and 3′3′-diaminobenzidine tetrahydrochloride (DAB) substrate kit (SK-140, Vector Laboratories, Newark, CA, USA) was used for the visualization. For negative control, PBS was used instead of primary antibody, and no positive staining was observed in this control.

The immunohistochemical staining results were visualized using an Axio Imager 1.0 microscope (Carl Zeiss, Jena, Germany). Images were captured with a Canon A640 Digital Camera System (Canon U.S.A., Inc, Melville, NY, USA). ETS1 protein staining was performed by two independent researchers, and the results were determined as the average of both researchers’ scores. Since the presence of ETS1 was observed in both the nucleus and the cytoplasm of cells, the scoring results were presented separately for the cellular compartments. The staining was evaluated by determining the staining intensity with the following scores: 1—weak intensity, 2—moderate intensity, and 3—strong intensity, as well as by determining the percentage of stained tumor cells (0–100%). The staining score for each compartment was calculated as the product of the staining intensity of that compartment and the percentage of stained cells/nuclei, while the overall staining score for each sample was calculated as the sum of the staining scores for both cell compartments (nucleus + cytoplasm).

### 4.7. Bioinformatic Analysis

Bioinformatic analysis was conducted to determine the sequence complementarity between miRs and ETS1, as well as to predict miR’s target scores. The MicroRNA Target Prediction Database (miRDB, http://www.mirdb.org accessed on 25 of May 2023) was used for miR target prediction.

### 4.8. Statistical Analysis

The distribution normality was assessed using the Shapiro-Wilk test for distribution type, along with visual methods such as Q-Q plots and histograms. All variables analyzed in this study followed non-Gaussian distributions, prompting the use of non-parametric tests for further analysis.

The related-samples Wilcoxon signed-rank test was applied for testing the difference between the median values of ETS1 between PTC and matched NMT and related-samples Friedman’s two-way analysis of variance by ranks was applied for testing the difference in the distribution of the values of the ETS1 in PTC and matched NMT. The Kruskal-Wallis Test was applied for comparing the distribution of the tested values among > 3 groups (PTC subtypes and NMT subtypes). The Median Test was applied for testing differences in median values if there were > 3 comparable groups (PTC subtypes and NMT subtypes). Pairwise Comparisons was applied when the Kruskal-Wallis or Median Test confirmed the existence of statistically significant differences among compared groups. Spearman’s correlation was applied for testing the correlation of ETS1 IHC staining and clinicopathological data of PTC patients. Receiver Operating Characteristic (ROC) analysis was performed to test if ETS1 could predict pT grade or lnm occurrence. The results were statistically significant at *p* < 0.05. Statistical analysis was carried out using SPSS 16.0 software (SPSS 16.0, Chicago, IL, USA).

## Figures and Tables

**Figure 1 ijms-26-01253-f001:**
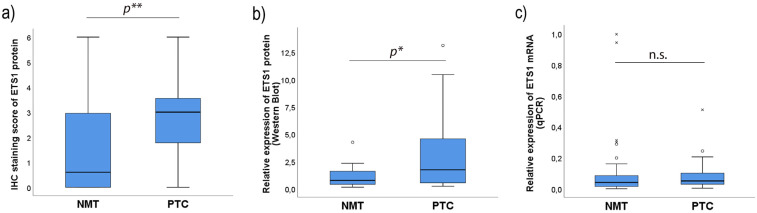
Expression of ETS1 in PTC and matched NMT: (**a**) ETS1 protein levels determined by IHC, (**b**) ETS1 protein levels determined by Western blot, (**c**) the level of ETS1 mRNA determined by qPCR; PTC: papillary thyroid carcinoma, NMT: matched nonmalignant thyroid tissue, IHC: immunohistochemistry, *p*: statistical significance (* *p* < 0.05, ** *p* < 0.01), n.s.: non-significant, *p* > 0.05. The results were statistically analyzed using the related-samples Wilcoxon signed rank test and related-samples Friedman’s two-way analysis of variance by ranks. The measurement data are expressed via box plots. The boxes represent the median value (the black horizontal line inside the box) and the interquartile range (upper and lower value); the symbols ‘o’ represent extreme values and the symbols ‘x’ represent outliers.

**Figure 2 ijms-26-01253-f002:**
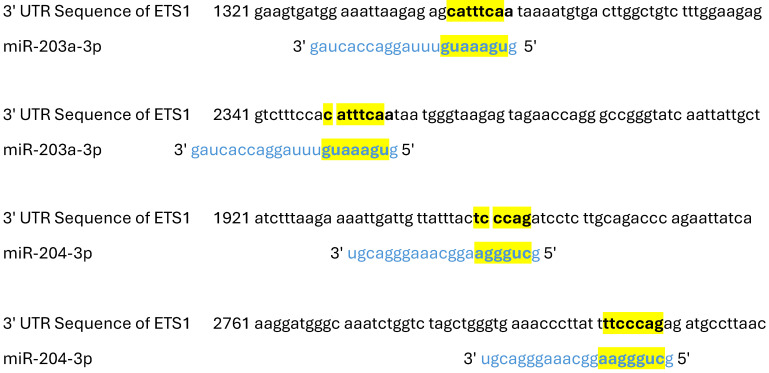
Positions of the complementary binding of ETS1 mRNA and miR-203a-3p/204-3p. 3′UTR- 3′ untranslated region of the ETS1 mRNA. The sequences colored with black letters are ETS1 mRNA, the sequences colored with blue letters are miRs, and the sequences highlighted in yellow are the complementary pairing of 3′ UTR mRNA with appropriate miR. The presented number is the position in the 3′ UTR of ETS1 mRNA.

**Figure 3 ijms-26-01253-f003:**
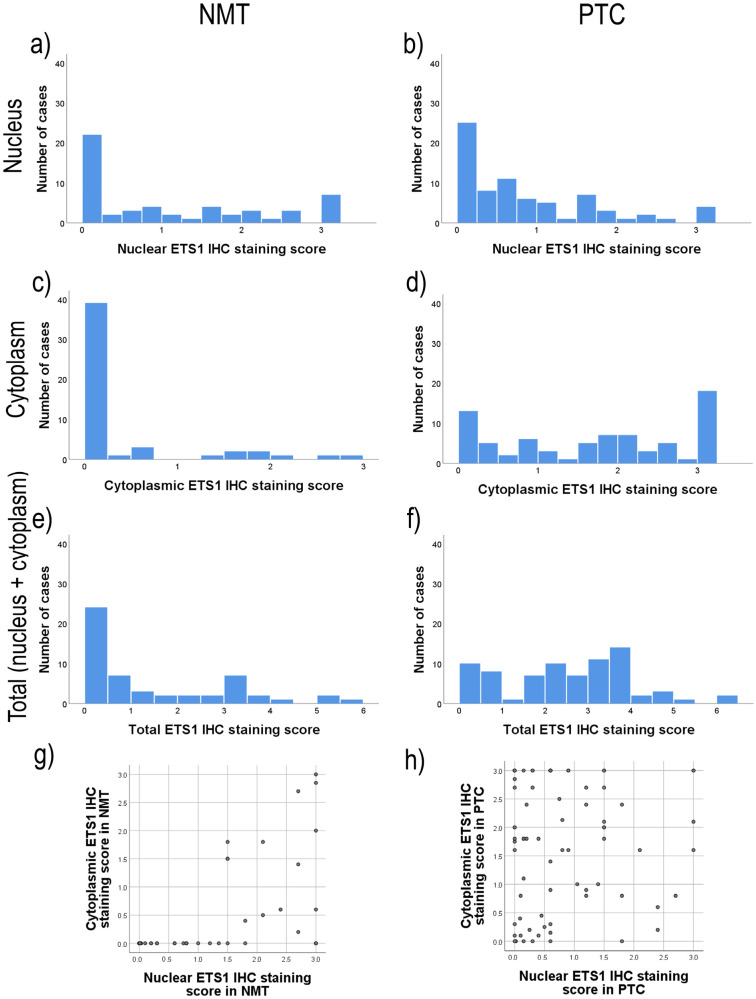
Distribution of ETS1 immunohistochemical staining by cell compartments in PTC and matched NMT. (**a**) nuclear ETS1 staining in NMT, (**b**) nuclear ETS1 staining in PTC, (**c**) cytoplasmic ETS1 staining in NMT, (**d**) cytoplasmic ETS1 staining in PTC, (**e**) total IHC staining of ETS1 in NMT, (**f**) total IHC staining of ETS1 in PTC, (**g**) correlation of nuclear and cytoplasmic ETS1 staining in NMT, (**h**) correlation of nuclear and cytoplasmic ETS1 staining in PTC. PTC: papillary thyroid carcinoma, NMT: nonmalignant thyroid tissue, IHC: immunohistochemistry.

**Figure 4 ijms-26-01253-f004:**
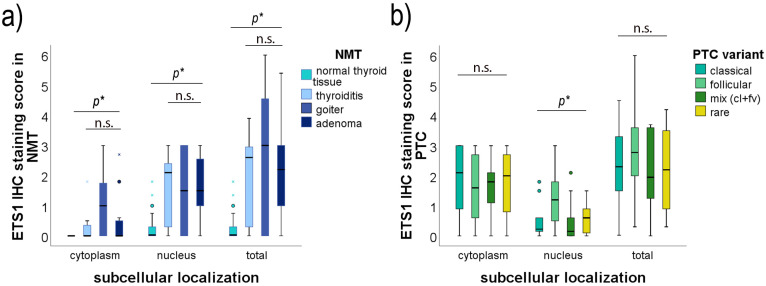
ETS1 IHC staining by cell compartments in (**a**) divergent nonmalignant thyroid tissues (NMT) and (**b**) divergent variants of papillary thyroid carcinoma (PTC). Mix is a mixture of classical (cl) and follicular variant (fv) of PTC. IHC: immunohistochemistry. *p*: statistical significance (* *p* < 0.05), n.s.: non-significant, *p* > 0.05. The results were statistically analyzed using the Kruskal-Wallis Test and the Median Test. The measurement data are expressed via box plots. The boxes represent the median value (the black horizontal line inside the box) and the interquartile range (upper and lower value); the symbols ‘o’ represent extreme values and the symbols ‘x’ represent outliers.

**Figure 5 ijms-26-01253-f005:**
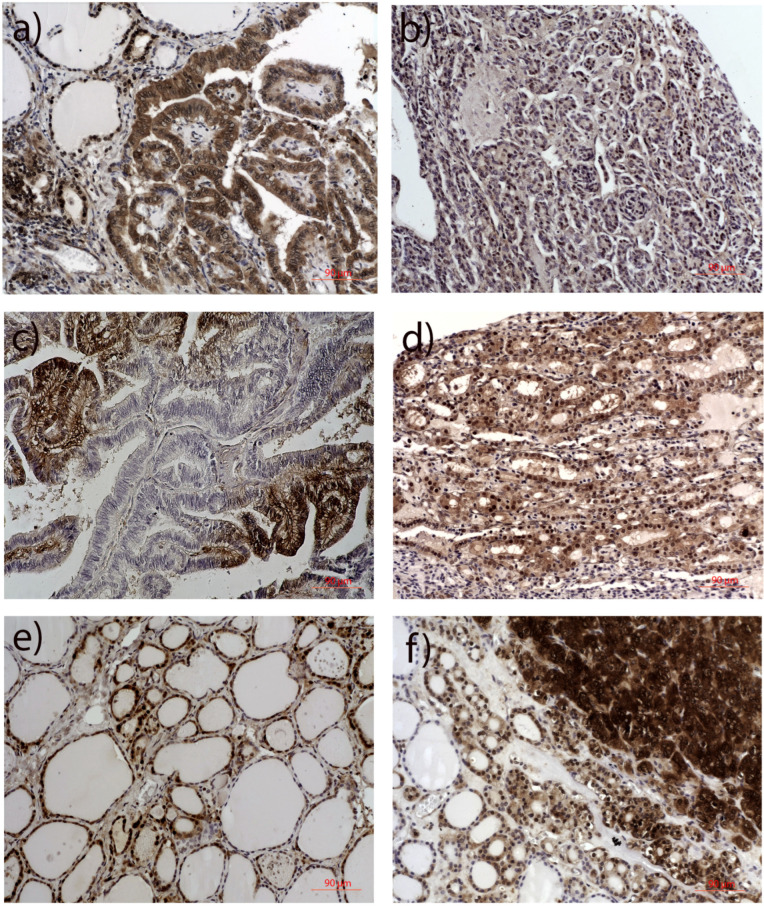
ETS1 IHC staining in PTC and adjacent NMT: (**a**) high cytoplasmic positivity of PTC cells and nuclear staining of surrounding NMT, (**b**) high nuclear with negative cytoplasmic staining of PTC cells, (**c**) heterogenic staining of PTC, (**d**) high nuclear with low cytoplasmic staining of PTC cells, (**e**) heterogenic staining of NMT, (**f**) heterogenic staining of divergent parts of PTC-high cytoplasmic staining of one PTC zone, high nuclear with low cytoplasmic staining of PTC cells close to NMT, and negative or low positive cytoplasmic staining of adjacent NMT. IHC: immunohistochemistry, PTC: papillary thyroid carcinoma, NMT: nonmalignant thyroid tissue. Original magnifications x10 in all.

**Figure 6 ijms-26-01253-f006:**
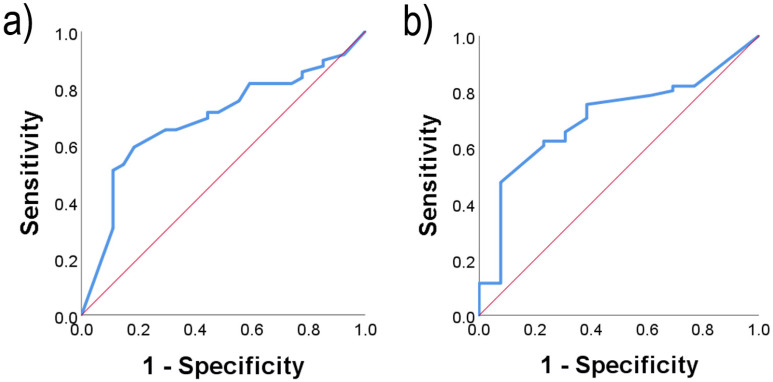
Receiver operating characteristic analysis for (**a**) the distinction of pT grade 1-2 vs. pT grade 3-4 according to the cytoplasmic levels of the ETS1 protein in PTC (AUC = 0.689, SE = 0.063, *p* = 0.007) and (**b**) the prediction of lymph node metastasis occurrence according to the immunohistochemical expression of ETS1 in the nucleus (AUC = 0.704, SE = 0.072, *p* = 0.021). PTC: papillary thyroid carcinoma. AUC: Area under the curve, SE: Standard error, *p*: statistical significance.

**Figure 7 ijms-26-01253-f007:**
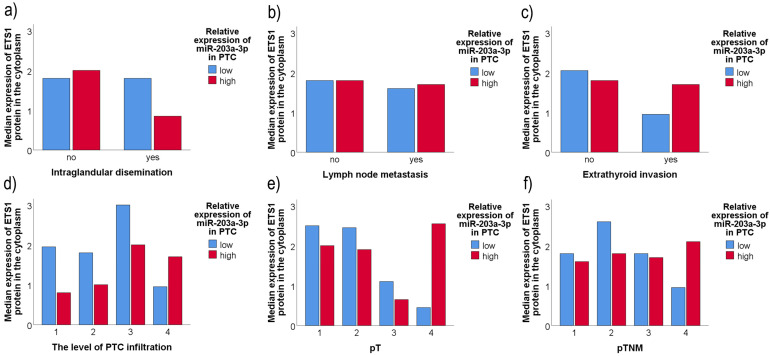
Relations between miR-203a-3p expression, median cytoplasmic levels of the ETS1 protein, and adverse clinicopathological findings in PTC patients. PTC: papillary thyroid carcinoma. ETS1 was determined by immunohistochemistry. MiR expression was determined by qPCR. (**a**) the influence of the ratio of the levels of ETS1 and miR-203a-3p expression on the development of intraglandular dissemination, (**b**) the influence of the ratio of the levels of ETS1 and miR-203a-3p expression on the development of metastasis in the regional lymph nodes, (**c**) the influence of the ratio of the levels of ETS1 and miR-203a-3p expression on the occurrence of invasion of thyroid capsule, (**d**) the influence of the ratio of the levels of ETS1 and miR-203a-3p expression on the depth of tumor infiltration through the gland, (**e**) the influence of the ratio of the levels of ETS1 and miR-203a-3p expression on the higher tumor pT status, (**f**) the influence of the ratio of the levels of ETS1 and miR-203a-3p expression on the higher pTNM stage of the patients.

**Table 1 ijms-26-01253-t001:** MicroRNA and target gene description.

NCBI Gene ID	2113	GenBank Accession	NM_001143820
Gene Symbol	*ETS1*	3′ UTR Length	3600
Gene Description	ETS proto-oncogene 1, transcription factor		
miRNA Name	miR-203a-3p	miRNA Sequence	GUGAAAUGUUUAGGACCACUAG
Target Score	63	Seed Location	1343, 2350
URL	https://mirdb.org/cgi-bin/target_detail.cgi?targetID=3441945		
miRNA Name	miR-204-3p	miRNA Sequence	GCUGGGAAGGCAAAGGGACGU
Target Score	98	Seed Location	1949, 2802
URL	https://mirdb.org/cgi-bin/target_detail.cgi?targetID=3259206		

**Table 2 ijms-26-01253-t002:** Correlation of ETS1 immunohistochemical staining by cell compartments with the occurrence of unfavorable clinicopathological parameters of PTC patients.

Clinicopathological Characteristics of PTC Patients	Cytoplasm		Nucleus	
	r	*p*-Value	r	*p*-Value
Gender	−0.062	0.597	−0.034	0.774
Age	−0.077	0.508	0.175	0.137
Tumor size	−0.175	0.131	0.141	0.230
Ei	−0.153	0.187	−0.067	0.571
ID	−0.149	0.197	0.188	0.108
LNM	−0.030	0.794	−0.271	0.020*
DTI	−0.052	0.658	−0.164	0.163
pT	−0.267	0.020 *	0.040	0.735
pTNM	−0.075	0.519	0.105	0.376

Ei: extrathyroid invasion, ID: intraglandular dissemination, LNM: lymph node metastasis, DTI: degree of tumor infiltration, pT: pT grade, pTNM: pTNM stage (for details see Section 4). r: correlation coefficient, *p*-value: statistical significance: * *p* < 0.05. Statistical analysis: Spearman’s correlation. PTC: papillary thyroid carcinoma.

## Data Availability

The data presented in this study are available on request from the corresponding author.

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
