# Peer review of "ETS1 Protein Expression May Be Altered by the Complementarity of ETS1 mRNA Sequences with miR-203a-3p and miR-204-3p in Papillary Thyroid Carcinoma"

_ijms, 2025, doi:10.3390/ijms26031253_

Round 1
Reviewer 1 Report
Comments and Suggestions for Authors
In this manuscript, the authors explore the expression of the ETS1 gene and its potential regulation by two microRNAs in papillary thyroid carcinoma (PTC). While the topic is of interest, the results presented do not provide sufficient evidence to support the authors' conclusions. I recommend rejecting this version of the manuscript. Below are my detailed comments:
--The quality of the figures needs significant improvement. The authors could adjust each figure to make them more professional and readable.
--Please prove more evidence that these two microRNAs could directly regulate ETS1 gene. More bench work should be conducted for this one (like knockdown or mimic of microRNAs).
--The rationale for focusing on the ETS1 gene in PTC patients is not clearly justified. Are there any previous laboratory findings that suggest ETS1 plays a significant role in the progression of PTC?
Author Response
Answer to the Reviewer 1:
Dear Reviewer,
We value the time and effort you have dedicated to reviewing our manuscript. We are glad that you have found the topic of our manuscript interesting and thank you for your valuable feedback and keen observation. In response to your suggestions, we have revised the manuscript. Before all, all the figures are improved, the conclusion and the title of the manuscript have been changed, and lines explaining ETS1's role in PTC development have been inserted. Please find a detailed answer for each of your comments:
- The quality of the figures needs significant improvement. The authors could adjust each figure to make them more professional and readable.
We would like to thank the reviewer for this insightful comment. We have made changes to all the figures in the manuscript, along with legends of each of them. Before all, at each figure/graph, the numbers and letters are scaled up, the resolution is raised, and in the legend of each figure, the p-value is added, the statistical test applied is named, and all the abbreviations and symbols are explained. Additionally, an appropriate p-value is added to Figure 1 and Figure 4, and a scale bar is inserted into the micrographs presented in Figure 5. Text highlighted in green indicates added or modified lines according to your suggestions:
Lines 124 – 129:
p-value – statistical significance: p* < 0.05, p** < 0.01, n.s.: non-significant, p > 0.05. The results were statistically analyzed using the Related-Samples Wilcoxon Signed Rank Test and Related-Samples Friedman's Two-Way Analysis of Variance by Ranks. The measurement data are expressed via box plots. The boxes represent the median value (the black horizontal line inside the box) and the interquartile range (upper and lower value); circles (o) represent extreme values, and whiskers (x) represent outliers.
Lines 167 – 171:
3’UTR- 3’ untranslated region of the ETS1 mRNA. The sequences colored with black letters are ETS1 mRNA, the sequences colored with blue letters are miRs, and the sequences highlighted in yellow are the complementary pairing of 3’UTR mRNA with appropriate miR. The presented number is the position in the 3’UTR of ETS1 mRNA.
Lines 218 -223:
Mix is a mixture of classical (cl) and follicular variant (fv) of PTC. IHC: immunohistochemistry. p-value – statistical significance: p* < 0.05, p** < 0.01, n.s.: non-significant, p > 0.05. The results were statistically analyzed using the Kruskal-Wallis Test and Median Test. The measurement data are expressed via box plots. The boxes represent the median value (the black horizontal line inside the box) and the interquartile range (upper and lower value); circles (o) represent extreme values, and whiskers (x) represent outliers.
Line 229: Original magnifications x10 in all.
Lines 240 – 241:
pT - pT grade, pTNM – pTNM stage (for details se the Material and Methods section).
Lines 253 – 258:
Receiver operating characteristic analysis for a) the distinction of pT grade 1-2 vs. pT grade 3-4 according to the cytoplasmic levels of ETS1 protein in PTC (AUC = 0.689, SE = 0.063, p = 0.007), and b) the prediction of lymph node metastasis occurrence according to the immunohystochemical expression of ETS1 in the nucleus (AUC = 0.704, SE = 0.072, p = 0.021). PTC: papillary thyroid carcinoma. AUC: Area under the curve, SE: Standard error, p: p-value – statistical significance: p* < 0.05, p** < 0.01.
Lines 268 – 278:
Relations between miR-203a-3p expression, median cytoplasmic levels of ETS1 protein, and adverse clinicopathological findings in PTC patients. PTC: papillary thyroid carcinoma. ETS1 was determined by immunohistochemistry. MiRs’ expression was determined by qPCR. a) the influence of the ratio of the levels of ETS1 and miR-203a-3p expression on the development of intraglandular dissemination, b) the influence of the ratio of the levels of ETS1 and miR-203a-3p expression on the development of metastasis in the regional lymph nodes, c) the influence of the ratio of the levels of ETS1 and miR-203a-3p expression on the occurrence of invasion of thyroid capsule, d) the influence of the ratio of the levels of ETS1 and miR-203a-3p expression on the depth of tumor infiltration through the gland, e) the influence of the ratio of the levels of ETS1 and miR-203a-3p expression on the higher tumor pT status, f) the influence of the ratio of the levels of ETS1 and miR-203a-3p expression on the higher pTNM stage of the patients
We sincerely hope that the figures in the revised version of our manuscript are sufficiently improved and readable.
- Please prove more evidence that these two microRNAs could directly regulate ETS1 gene. More bench work should be conducted for this one (like knockdown or mimic of microRNAs).
We sincerely understand the reviewer’s concern about the evidence of the influence of tested miRs on ETS1 expression. It is clear that our manuscript would be much improved with cell culture experiments with introducing knockdown or mimic of tested miRs. This kind of manipulation would really upgrade our results and strengthen our statement about the regulatory role of examined molecules, but unfortunately, we could not perform such experiments, not only for the lack of time but because we do not have adequate mimic/knockdown models at the present moment. Although we are not able to send the results of this experiment, we honestly think that the results obtained so far possess capacity for the assumption that the complementarity of the sequences of tested miRs with 3’UTR of ETS1 mRNA may influence ETS1 protein expression. The next project plan will surely incorporate such analyses. As we sincerely comprehend your apprehension, therefore, in the revised version of the manuscript we have revised the conclusions, better pointed out the limitations of the study, and changed the title of the manuscript:
Lines 36 – 37:
PTC aggression could be influenced by increased cytoplasmic ETS1 protein levels, which may be affected by reduced levels of miR-203a-3p or miR-204-3p.
Lines 108 – 111:
Understanding the molecular mechanisms by which these miRs influence the expression of the ETS1 protein may help identify new therapeutic targets for the management of PTC, especially in cases where traditional treatments are ineffective.
Lines 441-444:
Despite several limitations in our study, including a lack of cell culture experimental confirmation that miR-203a-3p and miR-204-3p could directly regulate ETS1 protein expression, a relatively small sample size, and no follow-up data, it is reasonable to believe that ETS1 activity is context dependent.
Lines 2-4:
ETS1 protein expression may be altered by the complementarity of ETS1 mRNA sequences with miR-203a-3p and miR-204-3p in papillary thyroid carcinoma
- The rationale for focusing on the ETS1 gene in PTC patients is not clearly justified. Are there any previous laboratory findings that suggest ETS1 plays a significant role in the progression of PTC?
We greatly appreciate the suggestion, as it helped us improve the text of the manuscript and clearly point out the value of the presented results. The explanation is now introduced in the introduction section of the manuscript (lines 65 -86) and in the discussion section of the manuscript (lines 317 – 324). Additionally, eight references are added to the manuscript that are listed in the References list (ref. num. 15-22, in lines 664 – 684).
Lines 65-86:
Although aberrant ETS1 activation has been documented in various solid tumors [5-12], a thorough and systematic analysis elucidating the role of ETS1 in thyroid tumor progression remains limited. Its expression in PTC is poorly characterized, and the clinical significance for PTC patients is ambiguous. While Fuhrer et al. [13] found elevated levels of ETS1 in PTC when compared to benign thyroid nodules and normal thyroid tissues, Nakayama et al. [14] observed strong expression of ETS1 in the majority of malignant thyroid tumors, a minority of benign thyroid tumors, and no expression in normal thyroid follicular cells. The TCGA study revealed ETS1 as a differentially expressed gene between lower and higher stages of PTC patients [15], and the transcriptional activity of ETS1 and ETS2 was found to be crucial for the transformation of thyroid follicular cells [16]. ETS1 mRNA expression was found to be upregulated in PTCs harboring the oncogene BRAFV600E mutation [17, 18], and mutation itself caused an increase in ETS1 protein expression and its nuclear accumulation [17]. Subsequent investigation revealed the direct role of ETS1 as a transcriptional regulator of toll-like receptor 4 (TLR4) gene expression downstream to MAPK/ERK signaling that is triggered by the presence of the BRAFV600E mutation [17]. Furthermore, the study of Song et al. [19] provided transcriptomic insights that the interaction between the BRAFV600E and TERT promoter mutations is mediated by ETS factors in PTC. Additionally, ETS1 was found to be a genome-wide effector of RAS/ERK signaling in epithelial cells [20]. Conversely, overexpression of the lncRNA SLC26A4-AS1 enhanced autophagy mediated by ITPR1 (the inositol 1,4,5-trisphosphate receptor type 1) and prevented the growth and progression of PTC by recruiting ETS1 [21, 22].
Lines 317 – 324:
However, Peyret et al.'s investigation [17] observed that although ETS1 was present in both the cytoplasm and the nucleus, the BRAFV600E mutation led to increased ETS1 protein expression and accumulation in the nucleus of thyroid cells that had undergone malignant transformation. Song et al. [19] have proposed a possible mechanism for the phenomenon of the synergistic effects of the two most common mutations in thyroid carcinomas on the poor clinical outcomes in PTC. They discovered that ETS factors induced by BRAFV600E selectively bound to the mutant TERT promoter, which in turn affects PTC invasiveness and progression in a synergistic manner.
Lines 664 – 684:
- Xu, Y., Gao, J., Wang, N., Zedenius, J., Nilsson, I. L., Lui, W. O., Xu, D., Juhlin, C. C., Larsson, C., & Mu, N. BRAF-induced EHF expression affects TERT in aggressive papillary thyroid cancer. J. Clin. Endocrinol. Metab. 2024, Advance online publication. https://doi.org/10.1210/clinem/dgae589.
- de Nigris, F.; Mega, T.; Berger, N.; Barone, M.V.; Santoro, M.; Viglietto, G.; Verde, P.; Fusco, A. Induction of ETS-1 and ETS-2 transcription factors is required for thyroid cell transformation. Cancer Res. 2001, 61, 2267–2275.
- Peyret, V., Nazar, M., Martín, M., Quintar, A. A., Fernandez, E. A., Geysels, R. C., Fuziwara, C. S., Montesinos, M. M., Mal-donado, C. A., Santisteban, P., Kimura, E. T., Pellizas, C. G., Nicola, J. P., & Masini-Repiso, A. M. Functional Toll-like Receptor 4 Overexpression in Papillary Thyroid Cancer by MAPK/ERK-Induced ETS1 Transcriptional Activity. Mol. Cancer Res. 2018, 16(5), 833–845. https://doi.org/10.1158/1541-7786.MCR-17-0433.
- Kim Y.H.; Choi Y.W.; Han J.H.; Lee J.; Soh E.Y.; Park S.H., et al. TSH signaling overcomes B-RafV600E-induced senescence in papillary thyroid carcinogenesis through regulation of DUSP6. Neoplasia 2014, 16, 1107–1120.
- Song, Y. S.; Yoo, S. K.; Kim, H. H.; Jung, G.; Oh, A. R.; Cha, J. Y.; et al. Interaction of BRAF-induced ETS factors with mutant TERT promoter in papillary thyroid cancer. Endocr. -Relat. Cancer 2019, 26(6), 629–641. https://doi.org/10.1530/ERC-17-0562.
- Plotnik, J.P.; Budka, J.A.; Ferris, M.W.; Hollenhorst, P.C. ETS1 is a genome-wide effector of RAS/ERK signaling in epithelial cells. Nucleic Acids Res. 2014, 42, 11928–11940.
- Peng, D.; Li, W.; Zhang, B.; Liu, X. Overexpression of lncRNA SLC26A4-AS1 Inhibits Papillary Thyroid Carcinoma Progression Through Recruiting ETS1 to Promote ITPR1-Mediated Autophagy. J. Cell. Mol. Med. 2021, 25(17), 8148–8158. https://doi.org/10.1111/jcmm.16545.
- Peng, D.; Li, W.; Zhang, B.; Liu, X. Overexpression of lncRNA SLC26A4-AS1 Inhibits Papillary Thyroid Carcinoma Progression Through Recruiting ETS1 to Promote ITPR1-Mediated Autophagy. J. Cell. Mol. Med. 2022, 26(9), 2750–2751. https://doi.org/10.1111/jcmm.17293.
Reviewer 2 Report
Comments and Suggestions for Authors
Dear authors,
The manuscript provides investigations into the role of ETS1 as prognostic marker in papillary thyroid carcinoma. This is a relevant area of research, particularly considering the increasing interest in miRNA as potential biomarkers in various diseases. Overall, this paper has a good quality. It has an interesting subject. The manuscript is well written in each part and easily accessible for readers. The choice of experiments was in line with the purposes. The results are supported by experimental techniques, by clear images. It’s my opinion this manuscript is suitable for the publication after a minor revision.
1. The introduction is well written, presenting an informative background information about papillary thyroid carcinoma (PTC) and its clinical management. The introduction spends considerable words on ETS1 and specific miRNAs. The objective was well described. I suggest a more detailed introduction of ETS1 mechanisms would help readers better understand the significance of studying ETS1 in this context.
2. The results are clear. However, figures have statistical significative symbols that were not reported in the caption. Please add (p.value …..)
Author Response
Thank you very much for your time, evaluation, and compliments. We appreciate your review of our paper and the useful insights you have shared. We would like to acknowledge the constructive criticisms and suggestions that, we believe, have contributed to improving the content and the style of the manuscript. In response to your suggestion, we have revised the manuscript according to your suggestions, which are highlighted in green inside the text:
- The introduction is well written, presenting an informative background information about papillary thyroid carcinoma (PTC) and its clinical management. The introduction spends considerable words on ETS1 and specific miRNAs. The objective was well described. I suggest a more detailed introduction of ETS1 mechanisms would help readers better understand the significance of studying ETS1 in this context.
We appreciate the reviewer's feedback because it helped us strengthen the organization of our work, clarify the acquired data, and emphasize the significance of the obtained results. According to the suggestion, more detailed ETS1 mechanisms and previously published data of ETS1 expression in PTC are inserted in the introduction (lines 65 -86) and discussion section (lines 317-324) of the manuscript. In addition, eight references are added to the paper and cited in the References section (ref. num. 15-22, in lines 664 – 684).
Lines 65-86:
Although aberrant ETS1 activation has been documented in various solid tumors [5-12], a thorough and systematic analysis elucidating the role of ETS1 in thyroid tumor progression remains limited. Its expression in PTC is poorly characterized, and the clinical significance for PTC patients is ambiguous. While Fuhrer et al. [13] found elevated levels of ETS1 in PTC when compared to benign thyroid nodules and normal thyroid tissues, Nakayama et al. [14] observed strong expression of ETS1 in the majority of malignant thyroid tumors, a minority of benign thyroid tumors, and no expression in normal thyroid follicular cells. The TCGA study revealed ETS1 as a differentially expressed gene between lower and higher stages of PTC patients [15], and the transcriptional activity of ETS1 and ETS2 was found to be crucial for the transformation of thyroid follicular cells [16]. ETS1 mRNA expression was found to be upregulated in PTCs harboring the oncogene BRAFV600E mutation [17, 18], and mutation itself caused an increase in ETS1 protein expression and its nuclear accumulation [17]. Subsequent investigation revealed the direct role of ETS1 as a transcriptional regulator of toll-like receptor 4 (TLR4) gene expression downstream to MAPK/ERK signaling that is triggered by the presence of the BRAFV600E mutation [17]. Furthermore, the study of Song et al. [19] provided transcriptomic insights that the interaction between the BRAFV600E and TERT promoter mutations is mediated by ETS factors in PTC. Additionally, ETS1 was found to be a genome-wide effector of RAS/ERK signaling in epithelial cells [20]. Conversely, overexpression of the lncRNA SLC26A4-AS1 enhanced autophagy mediated by ITPR1 (the inositol 1,4,5-trisphosphate receptor type 1) and prevented the growth and progression of PTC by recruiting ETS1 [21, 22].
Lines 317 – 324:
However, Peyret et al.'s investigation [17] observed that although ETS1 was present in both the cytoplasm and the nucleus, the BRAFV600E mutation led to increased ETS1 protein expression and accumulation in the nucleus of thyroid cells that had undergone malignant transformation. Song et al. [19] have proposed a possible mechanism for the phenomenon of the synergistic effects of the two most common mutations in thyroid carcinomas on the poor clinical outcomes in PTC. They discovered that ETS factors induced by BRAFV600E selectively bound to the mutant TERT promoter, which in turn affects PTC invasiveness and progression in a synergistic manner.
Lines 664 – 684:
- Xu, Y., Gao, J., Wang, N., Zedenius, J., Nilsson, I. L., Lui, W. O., Xu, D., Juhlin, C. C., Larsson, C., & Mu, N. BRAF-induced EHF expression affects TERT in aggressive papillary thyroid cancer. J. Clin. Endocrinol. Metab. 2024, Advance online publication. https://doi.org/10.1210/clinem/dgae589.
- de Nigris, F.; Mega, T.; Berger, N.; Barone, M.V.; Santoro, M.; Viglietto, G.; Verde, P.; Fusco, A. Induction of ETS-1 and ETS-2 transcription factors is required for thyroid cell transformation. Cancer Res. 2001, 61, 2267–2275.
- Peyret, V., Nazar, M., Martín, M., Quintar, A. A., Fernandez, E. A., Geysels, R. C., Fuziwara, C. S., Montesinos, M. M., Mal-donado, C. A., Santisteban, P., Kimura, E. T., Pellizas, C. G., Nicola, J. P., & Masini-Repiso, A. M. Functional Toll-like Receptor 4 Overexpression in Papillary Thyroid Cancer by MAPK/ERK-Induced ETS1 Transcriptional Activity. Mol. Cancer Res. 2018, 16(5), 833–845. https://doi.org/10.1158/1541-7786.MCR-17-0433.
- Kim Y.H.; Choi Y.W.; Han J.H.; Lee J.; Soh E.Y.; Park S.H., et al. TSH signaling overcomes B-RafV600E-induced senescence in papillary thyroid carcinogenesis through regulation of DUSP6. Neoplasia 2014, 16, 1107–1120.
- Song, Y. S.; Yoo, S. K.; Kim, H. H.; Jung, G.; Oh, A. R.; Cha, J. Y.; et al. Interaction of BRAF-induced ETS factors with mutant TERT promoter in papillary thyroid cancer. Endocr. -Relat. Cancer 2019, 26(6), 629–641. https://doi.org/10.1530/ERC-17-0562.
- Plotnik, J.P.; Budka, J.A.; Ferris, M.W.; Hollenhorst, P.C. ETS1 is a genome-wide effector of RAS/ERK signaling in epithelial cells. Nucleic Acids Res. 2014, 42, 11928–11940.
- Peng, D.; Li, W.; Zhang, B.; Liu, X. Overexpression of lncRNA SLC26A4-AS1 Inhibits Papillary Thyroid Carcinoma Progression Through Recruiting ETS1 to Promote ITPR1-Mediated Autophagy. J. Cell. Mol. Med. 2021, 25(17), 8148–8158. https://doi.org/10.1111/jcmm.16545.
- Peng, D.; Li, W.; Zhang, B.; Liu, X. Overexpression of lncRNA SLC26A4-AS1 Inhibits Papillary Thyroid Carcinoma Progression Through Recruiting ETS1 to Promote ITPR1-Mediated Autophagy. J. Cell. Mol. Med. 2022, 26(9), 2750–2751. https://doi.org/10.1111/jcmm.17293.
- The results are clear. However, figures have statistical significative symbols that were not reported in the caption. Please add (p.value …..)
Thank you for this suggestion. We have made changes to all the figures in the manuscript, along with legends of each of them. Before all, at each figure/graph, the numbers and letters are scaled up, the resolution is raised, and in the legend of each figure, the p-value is added, the statistical test applied is named, and all the abbreviations and symbols are explained. Additionally, an appropriate p-value is added to Figure 1 and Figure 4, and a scale bar is inserted into the micrographs presented in Figure 5. Text highlighted in green indicates added or modified lines according to your suggestions: lines 124 – 129, lines 167 – 171, lines 218 -223, line 229, lines 240 - 241, lines 253 – 258, and lines 268 – 278. We sincerely hope that the figures in the revised version of our manuscript are sufficiently improved and readable.
Lines 124 – 129:
p-value – statistical significance: p* < 0.05, p** < 0.01, n.s.: non-significant, p > 0.05. The results were statistically analyzed using the Related-Samples Wilcoxon Signed Rank Test and Related-Samples Friedman's Two-Way Analysis of Variance by Ranks. The measurement data are expressed via box plots. The boxes represent the median value (the black horizontal line inside the box) and the interquartile range (upper and lower value); circles (o) represent extreme values, and whiskers (x) represent outliers.
Lines 167 – 171:
3’UTR- 3’ untranslated region of the ETS1 mRNA. The sequences colored with black letters are ETS1 mRNA, the sequences colored with blue letters are miRs, and the sequences highlighted in yellow are the complementary pairing of 3’UTR mRNA with appropriate miR. The presented number is the position in the 3’UTR of ETS1 mRNA.
Lines 218 -223:
Mix is a mixture of classical (cl) and follicular variant (fv) of PTC. IHC: immunohistochemistry. p-value – statistical significance: p* < 0.05, p** < 0.01, n.s.: non-significant, p > 0.05. The results were statistically analyzed using the Kruskal-Wallis Test and Median Test. The measurement data are expressed via box plots. The boxes represent the median value (the black horizontal line inside the box) and the interquartile range (upper and lower value); circles (o) represent extreme values, and whiskers (x) represent outliers.
Line 229: Original magnifications x10 in all.
Lines 240 – 241:
pT - pT grade, pTNM – pTNM stage (for details se the Material and Methods section).
Lines 253 – 258:
Receiver operating characteristic analysis for a) the distinction of pT grade 1-2 vs. pT grade 3-4 according to the cytoplasmic levels of ETS1 protein in PTC (AUC = 0.689, SE = 0.063, p = 0.007), and b) the prediction of lymph node metastasis occurrence according to the immunohystochemical expression of ETS1 in the nucleus (AUC = 0.704, SE = 0.072, p = 0.021). PTC: papillary thyroid carcinoma. AUC: Area under the curve, SE: Standard error, p: p-value – statistical significance: p* < 0.05, p** < 0.01.
Lines 268 – 278:
Relations between miR-203a-3p expression, median cytoplasmic levels of ETS1 protein, and adverse clinicopathological findings in PTC patients. PTC: papillary thyroid carcinoma. ETS1 was determined by immunohistochemistry. MiRs’ expression was determined by qPCR. a) the influence of the ratio of the levels of ETS1 and miR-203a-3p expression on the development of intraglandular dissemination, b) the influence of the ratio of the levels of ETS1 and miR-203a-3p expression on the development of metastasis in the regional lymph nodes, c) the influence of the ratio of the levels of ETS1 and miR-203a-3p expression on the occurrence of invasion of thyroid capsule, d) the influence of the ratio of the levels of ETS1 and miR-203a-3p expression on the depth of tumor infiltration through the gland, e) the influence of the ratio of the levels of ETS1 and miR-203a-3p expression on the higher tumor pT status, f) the influence of the ratio of the levels of ETS1 and miR-203a-3p expression on the higher pTNM stage of the patients